# Current Control Method of Vehicle Permanent Magnet Synchronous Motor Based on Active Disturbance Rejection Control

**Jinyu Wang [1], Qiang Miao [2], Xiaomin Zhou [1,\*], Lipeng Sun [2], Dawei Gao [3] and Haifeng Lu [4]**

[1] School of Mechanical Engineering, University of Science and Technology Beijing, Beijing 100083, China
[2] Electric Control Research Institute, Weichai Power Co., Ltd., Weifang 261061, China
[3] School of Vehicle and Mobility, Tsinghua University, Beijing 100084, China
[4] Department of Electrical Engineering, Tsinghua University, Beijing 100084, China
**\*** Correspondence: zhouxiaomin@ustb.edu.cn

**Abstract:** Due to the frequently changing working conditions and complex operating environment of electric vehicle permanent magnet synchronous motor(PMSM), the motor parameters change dramatically. However, the performance of the PI current regulator, which is the most widely used, is sensitive to motor parameters and has weak robustness, which will lead to the deterioration of motor control system performance. To address this problem, active disturbance rejection control (ADRC) technology is applied to the PMSM current loop control. Firstly, the traditional ADRC current regulator is designed, and the performance and parameter tuning laws of the extended state observer are analyzed by the method of frequency domain analysis. Then, the traditional ADRC algorithm is improved in three aspects: observation error compensation, utilization of model information and anti-windup. After that, simulations and bench test validation are performed. The simulation results show that the improved ADRC current regulator is more robust in the face of parameter changes. The torque step test results show that the improved ADRC current regulator has fast dynamic response without overshoot and has high robustness when the motor parameters change. The dynamic test results show that the improved ADRC current regulator has high robustness when the load, speed and motor parameters change, and the anti-windup measures designed can effectively overcome the integral saturation phenomenon.

**Keywords:** electric vehicle; permanent magnet synchronous motor; active disturbance rejection control; current loop; robustness

## 1. Introduction

Because of its good mechanical characteristics, high power density and high reliability, permanent magnet synchronous motor (PMSM) is widely used in the field of electric vehicle drive motor [1,2]. The performance of the drive motor system plays a critical role in the performance of the entire vehicle. With the development of modern control theory, many advanced control technologies are used in PMSM control, such as model predictive control (MPC) [3], internal model control (IMC) [4], model reference adaptive system (MRAS) [5], etc. These control methods can improve the control accuracy and dynamic performance of the motor control algorithm to a certain extent. However, so far, these advanced algorithms have not been widely used in vehicle motors due to their sensitivity to parameters, computational complexity and tuning complexity. Vector control is still the mainstream of the vehicle PMSM control algorithm. In vector control, the current regulator, as the innermost loop regulator of the motor control system, directly affects the performance of the drive motor system. The most commonly used current regulator is the PI regulator, whose parameters are directly related to motor stator inductance and resistance. Frequent changes in operating conditions and temperature during vehicle

driving will cause changes in parameters, such as stator resistance, flux linkage and stator inductance, which will cause PI current regulator parameters to mismatch and lead to degradation of control performance [6–8].

The active disturbance rejection control(ADRC) is based on the idea of disturbance observation. The uncertain factors, such as parameter changes and load disturbances in the system, are attributed to total disturbances, which are observed and compensated by the extended state observer (ESO) [9,10]. It is very suitable for the PMSM control system of electric vehicles. At present, there are many precedents of applying ADRC technology to PMSM current loop control; for example, Lin, P. [11] designed linear-nonlinear ADRC regulators for PMSM speed loop and current loop regulators, and verified the robustness of ADRC regulators through a bench test, however, the proposed algorithm has many parameters and is difficult to tune, and the nonlinear functions contained in the algorithm put forward higher requirements for hardware computing capability Zhao, R. [12] designed ADRC current regulator with differential predictor. The simulation results show that its dynamic performance and robustness are better than the PI regulator, however, the robustness of the proposed algorithm when the motor parameters change was not analyzed and verified, and the algorithm was not bench tested. Tian, M. [13] combined ADRC with repetitive control and verified the suppression effect of this method on AC and DC interference through a bench test, and the robustness to inductance parameter mismatch was verified by changing the compensation factors ($b_d$ and $b_q$). Ren, L. [14] designed a PMSM sliding mode active disturbance rejection current regulator by combining ADRC and sliding mode control. The simulation results show that it can improve the disturbance rejection ability of the system and reduce chattering, but it also does not involve the change of motor parameters. The above scholars have verified the performance of ADRC through a variety of bench test or simulations; but at present, there is little research on the robustness of ADRC when the motor parameters change.

Firstly, based on the PMSM mathematical model, the first-order linear ADRC current regulator is designed and improved. The improvement involves three aspects: observation error compensation, utilization of model information and anti-windup; then, the robustness of ADRC current regulator is verified by simulation when the stator inductance changes. Finally, the step response performance and the performance under high dynamic conditions are verified by bench tests.

## 2. Design and Improvement of ADRC Current Regulator

### 2.1. Traditional ADRC Current Regulator

To simplify the analysis, factors such as magnetic circuit saturation, motor loss and manufacturing process are ignored. The transient-state dq-axis voltage equations of a PMSM are expressed below [15]:

$$\begin{cases} u_d = L_d \dot{i}_d + R_s i_d - \omega_e L_q i_q + w_d \\ u_q = L_q \dot{i}_q + R_s i_q + \omega_e L_d i_d + \omega_e \psi_f + w_q \end{cases} \tag{1}$$

where $i_d$ and $i_q$ are dq-axis currents; $L_d$ and $L_q$ are dq-axis inductances; $u_d$ and $u_q$ are dq-axis voltages; $\omega_e$ is the electrical angular speed; $\psi_f$ is the rotor flux linkage; $R_s$ is the stator resistance; $w_d$ and $w_q$ are unmodeled dynamics.

The back EMF, stator resistance voltage, cross coupling term and unmodeled dynamics are combined into total disturbances, which can be expressed as:

$$\begin{cases} f_d = -\frac{R_s i_d}{L_d} + \frac{\omega_e L_q i_q}{L_d} - \frac{w_d}{L_d} \\ f_q = -\frac{R_s i_q}{L_q} - \frac{\omega_e L_d i_d}{L_q} - \frac{\omega_e \psi_f}{L_q} - \frac{w_q}{L_q} \end{cases} \tag{2}$$

In addition, make $b_d = 1/L_d$, $b_q = 1/L_q$, ($b_d$ and $b_q$ are called compensation factors), and expand the total disturbance into a new state variable; (1) can be written in the form of extended state as shown in (3):

$$\begin{cases} \dot{i}_d = f_d + b_d u_d \\ \dot{i}_q = f_q + b_q u_q \end{cases} \tag{3}$$

Similar $i_{d/q}$ and $u_{d/q}$ forms are used to express the physical quantities of d-axis or q-axis to simplify the analysis; (3) can be expressed as the matrix form shown in (4):

$$\begin{cases} \dot{x}_{d/q} = Ax_{d/q} + Bu_{d/q} \\ y_{d/q} = Cx_{d/q} \end{cases} \tag{4}$$

where $x_{d/q} = \begin{bmatrix} i_{d/q} & f_{d/q} \end{bmatrix}^T$, and the specific forms of $A$, $B$ and $C$ are shown in (5):

$$A = \begin{bmatrix} 0 & 1 \\ 0 & 0 \end{bmatrix}, B = \begin{bmatrix} b_{d/q} \\ 0 \end{bmatrix}, C = \begin{bmatrix} 1 & 0 \end{bmatrix} \tag{5}$$

The design of Luenberger observer based on (4) is shown in (6):

$$\dot{z}_{d/q} = (A - L_{d/q}C)z_{d/q} + Bu_{d/q} + L_{d/q}y_{d/q} \tag{6}$$

where $z_{d/q} = \begin{bmatrix} z_{d1/q1} & z_{d2/q2} \end{bmatrix}$ is the observed value of $x_{d/q}$, $z_{d1/q1}$ is the observed value of dq-axis current; $z_{d2/q2}$ is the observed value of dq-axis total disturbance; $L_{d/q} = \begin{bmatrix} \beta_{d1/q1} & \beta_{d2/q2} \end{bmatrix}^T$ is the gain of the observer. Finally, the specific form of linear ESO (LESO) is obtained, as shown in (7):

$$\begin{cases} e_{d1/q1} = z_{d1/q1} - i_{d/q} \\ \dot{z}_{d1/q1} = z_{d2/q2} - \beta_{d1/q1}e_{d1/q1} + b_{d/q}u_{d/q} \\ \dot{z}_{d2/q2} = -\beta_{d2/q2}e_{d1/q1} \end{cases} \tag{7}$$

The linear state error feedback(LSEF) can be expressed as a linear combination of errors of each order of state variables and then the total disturbance observed value $z_{d2/q2}$ is feedforward compensated. The specific form of LSEF can be expressed as:

$$u_{d/q} = \frac{(k_{d/q}(i_{d/q}^* - z_{d1/q1}) - z_{d2/q2})}{b_{d/q}} \tag{8}$$

where $i_{d/q}^*$ is the current demand value of dq-axis; $k_{d/q}$ is the proportional gain to be tuned. The block diagram of the ADRC current regulator is shown in Figure 1.

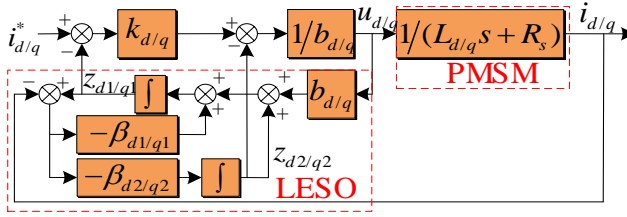

**Figure 1.** ADRC current regulator.

## 2.2. Observation Error Compensation

LESO observes and compensates the total disturbance of the system. In order to analyze the influence of the total disturbance on the observation accuracy of LESO, the observation errors of $i_{d/q}$ and $f_{d/q}$ are defined as:

$$\begin{cases} e_{d1/q1} = z_{d1/q1} - i_{d/q} \\ e_{d2/q2} = z_{d2/q2} - f_{d/q} \end{cases} \tag{9}$$

Based on (3) and (7), the differential equation of observation error can be expressed as:

$$\begin{cases} \dot{e}_{d1/q1} = e_{d2/q2} - \beta_{d1/q1}e_{d1/q1} \\ \dot{e}_{d2/q2} = -\beta_{d2/q2}e_{d1/q1} - \dot{f}_{d/q} \end{cases} \tag{10}$$

Laplace transform (10):

$$\begin{cases} e_{d1/q1}s^2 = -\beta_{d2/q2}e_{d1/q1} - f_{d/q}s - \beta_{d1/q1}e_{d1/q1}s \\ -(e_{d2/q2}s + f_{d/q}s)s = \beta_{d2/q2}e_{d2/q2} + \beta_{d1/q1}(e_{d2/q2}s + f_{d/q}s) \end{cases} \tag{11}$$

Arrange (11) to obtain the transfer function from the total disturbance to the observation error of each order of state variable as follows:

$$\begin{cases} G_{e_{d1/q1}}(s) = \dfrac{e_{d1/q1}(s)}{f_{d/q}(s)} = -\dfrac{s}{s^2 + \beta_{d1/q1}s + \beta_{d2/q2}} \\ G_{e_{d2/q2}}(s) = \dfrac{e_{d2/q2}(s)}{f_{d/q}(s)} = -\dfrac{s(s + \beta_{d1/q1})}{s^2 + \beta_{d1/q1}s + \beta_{d2/q2}} \end{cases} \tag{12}$$

Based on the bandwidth method proposed in the literature [16], the gains of LESO can be tuned. The characteristic equation of LESO can be easily obtained from (6), and the two poles of LESO are arranged at $-\omega_{d/q}$:

$$\det(\lambda I - (A - L_{d/q}C)) = \lambda^2 + \beta_{d1/q1}\lambda + \beta_{d2/q2} = \left(\lambda + \omega_{d/q}\right)^2 \tag{13}$$

where $I$ is a second-order identity matrix. According to (13):

$$\begin{cases} \beta_{d1/q1} = 2\omega_{d/q} \\ \beta_{d2/q2} = \omega_{d/q}^2 \end{cases} \tag{14}$$

$\omega_{d/q}$ becomes the only LESO parameter to be tuned, which is called observer bandwidth. According to literature [17], when $\omega_{d/q}$ is greater than 0, the system is stable, so $\omega_{d/q}$ should be a positive number. Changing the observer bandwidth, the frequency domain characteristics of (12) are shown in Figure 2a,b. Figure 2 shows that the larger the $\omega_{d/q}$ is, the smaller the observation error of LESO in the middle and the low frequency bands of $i_{d/q}$ and $f_{d/q}$, the higher the observation accuracy and the better the disturbance rejection performance. However, LESO cannot observe disturbances whose frequency is higher than its bandwidth very well, so its performance is limited by bandwidth. If the bandwidth is too large, the suppression ability of LESO to high-frequency noise will be reduced, so $\omega_{d/q}$ cannot be too large. Therefore, in the case of limited bandwidth, it is necessary to design the observation error compensation measures.

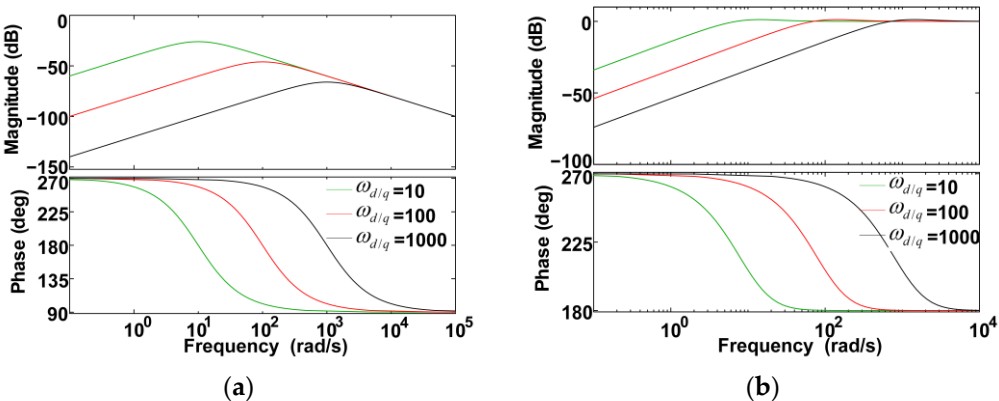

**Figure 2.** Bode diagram of observation error transfer function: (**a**) $G_{e_{d1/q1}}(s)$; (**b**) $G_{e_{d2/q2}}(s)$.

Substitute (8) into (3), and (3) can be expressed as:

$$\dot{i}_{d/q} = f_{d/q} + b_{d/q}u_{d/q} = f_{d/q} + b_{d/q}\frac{k_{d/q}(i^*_{d/q}-z_{d1/q1})-z_{d2/q2}}{b_{d/q}}$$
$$= k_{d/q}(i^*_{d/q} - z_{d1/q1}) - z_{d2/q2} + f_{d/q}$$

(15)

Considering the observation error shown in (9), (15) can be expressed as:

$$\dot{i}_{d/q} = (-k_{d/q}e_{d1/q1} - e_{d2/q2}) + k_{d/q}(i^*_{d/q} - i_{d/q})$$

(16)

The error term $\widetilde{E_{d/q}}$ outside the ideal system can be expressed as

$$\widetilde{E_{d/q}} = -k_{d/q}e_{d1/q1} - e_{d2/q2}$$

(17)

The observation error of the total disturbance cannot be calculated, so it needs to be replaced with a known quantity according to (10), and then (17) can be expressed as:

$$\widetilde{E_{d/q}} = -k_{d/q}e_{d1/q1} - \dot{e}_{d1/q1} - \beta_{d1/q1}e_{d1/q1}$$

(18)

When ignoring the differential term that is difficult to obtain, $\widetilde{E_{d/q}}$ can be expressed as:

$$\widetilde{E_{d/q}} = -k_{d/q}e_{d1/q1} - \beta_{d1/q1}e_{d1/q1}$$

(19)

Through feedforward, $\widetilde{E_{d/q}}$ is introduced into LSEF to offset the observation error, the LSEF with the observation error compensation function can be expressed as:

$$u_{d/q} = \frac{k_{d/q}(i^*_{d/q} - z_{d1/q1}) - z_{d2/q2} + (k_{d/q} + \beta_{d1/q1})e_{d1/q1}}{b_{d/q}}$$

(20)

### 2.3. Utilization of Model Information

As mentioned above, ADRC regards back EMF, cross coupling term and stator resistance voltage as total disturbances and relies on LESO for observation. Under high speed and large torque conditions, this total disturbance value will be very large, and relying solely on LESO for observation will slow the convergence speed of LESO, thus affecting the dynamic performance. The rated parameter values of PMSM are generally known, so the values of back EMF, cross coupling term and stator resistance voltage under rated parameters can be calculated. LESO only observes unmodeled dynamics and disturbances due to parameter changes [18].

LESO with known model information can be expressed as:

$$\begin{cases} e_{d1/q1} = z_{d1/q1} - i_{d/q} \\ \dot{z}_{d1/q1} = z_{d2/q2} - \beta_{d1/q1}e_{d1/q1} + b_{d/q}(u_{d/q} + f_{pd/pq}) \\ \dot{z}_{d2/q2} = -\beta_{d2/q2}e_{d1/q1} \end{cases} \quad (21)$$

where $f_{pd/pq}$ is the known model information part, which can be expressed as:

$$\begin{cases} f_{pq} = -\widetilde{R}_s i_q - \omega_e(\widetilde{\psi}_f + \widetilde{L}_d i_d) \\ f_{pd} = -\widetilde{R}_s i_d + \omega_e \widetilde{L}_q i_q \end{cases} \quad (22)$$

where $\widetilde{R}_s$, $\widetilde{\psi}_f$, $\widetilde{L}_d$ and $\widetilde{L}_q$ are the rated values of motor parameters, which can be obtained from the PMSM nameplate or technical manual. In addition, (22) can also be used as a feedforward decoupling of the current loop. LSEF with feedforward decoupling can be expressed as:

$$u_{d/q} = \frac{k_{d/q}(i^*_{d/q} - z_{d1/q1}) - z_{d2/q2} + (k_{d/q} + \beta_{d1/q1})e_{d1/q1}}{b_{d/q}} - f_{pd/pq} \quad (23)$$

*2.4. Anti-Windup*

In order to avoid voltage saturation caused by the regulator output exceeding the DC voltage limit, the current regulator output is usually limited. When PMSM operates at high speed and high torque, or DC voltage fluctuates, the output of the regulator may exceed the limit value. At this time, the current cannot reach the demand value, resulting in the continuous accumulation of the regulator's integral link and deterioration of the regulator's performance. This phenomenon is called integral saturation [19].

In order to avoid the harmful effect of integral saturation on the performance of the regulator, various anti-windup measures have been proposed and applied by researchers. There are three common anti-windup design methods: integral separation [20], limit weakening integral [21] and feedback suppression [22,23]. Among them, integral separation and limit weakening of integral will make LESO lose all regulation ability when integral saturation occurs, which is not suitable for ADRC. Feedback suppression introduces the error before and after amplitude limiting into the integral link through negative feedback. The integral link is used to eliminate this error and make the regulator exit from the saturation state. This anti-windup method introduces an anti-windup coefficient, which can adjust the strength of anti-windup effect according to needs. This anti-windup measure makes the regulator exit from integral saturation faster.

In this paper, the anti-windup measure of an ADRC current regulator is designed by the feedback suppression method. The LESO with anti-windup function can be expressed as:

$$\begin{cases} e_{d1/q1} = z_{d1/q1} - i_{d/q} - k_{cd/cq}(sat(u_{d/q}) - u_{d/q}) \\ \dot{z}_{d1/q1} = z_{d2/q2} - \beta_{d1/q1}e_{d1/q1} + b_{d/q}(u_{d/q} + f_{pd/pq}) \\ \dot{z}_{d2/q2} = -\beta_{d2/q2}e_{d1/q1} \end{cases} \quad (24)$$

where $k_{cd/cq}$ is anti-windup gain and $sat(u_{d/q})$ is output of the current regulator after amplitude limiting. The larger the $k_{cd/cq}$ value is, the stronger the anti-windup effect. However, the larger the $k_{cd/cq}$ value is, the more unstable will be the LESO. Therefore, it needs to be manually tuned to a suitable value in practical applications [24].

So far, the improved ADRC current regulator has been designed.

## 3. Simulation Analysis

### 3.1. Establishment of Simulink Model

In order to make the simulation closer to reality, a double simulation rate model is built to simulate and analyze the motor control system. The simulation step of the controlled object (inverter and PMSM) model is 1 MHz, and the simulation step of the control algorithm part is 5 kHz. The two parts are connected by a rate conversion module. The motor operates in the torque (current) closed-loop mode. Based on the mathematical model of PMSM, a PMSM model with variable parameters during simulation operation is built to simulate the changes of motor parameters in the actual system. The simulation control block diagram is shown in Figure 3. The motor parameters used for simulation are shown in Table 1.

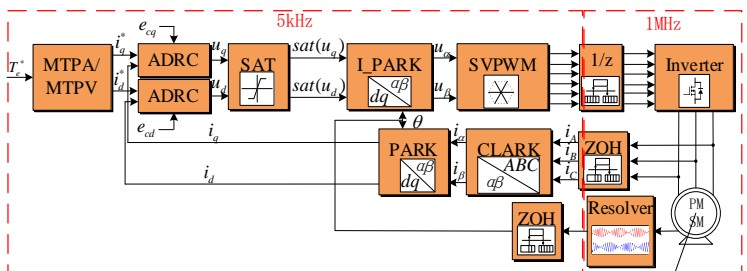

**Figure 3.** PMSM single closed loop control block diagram.

**Table 1.** Motor parameters.

| Parameter Name | Parameter Value |
|---|---|
| Number of phases | 3 |
| Rated DC voltage/V | 540 |
| Rated/peak power/kW | 130/260 |
| Rated/peak current/Arms | 230/525 |
| Rated/peak speed/(r·min$^{-1}$) | 1350/3000 |
| Rated/peak torque/N·m | 955/2800 |
| Rated d-axis inductance/mH | 0.618 |
| Rated q-axis inductance/mH | 0.197 |
| Flux linkage/Wb | 0.344 |
| Stator resistance/Ω | 0.035 |
| Number of pole pairs | 6 |

### 3.2. Analysis of Simulation Results

Due to the saturation and cross saturation effects, the stator inductances $L_d$ and $L_q$ change with the dq-axis current in a wide range, and the speed of change is very fast, which has a great impact on the system performance. Therefore, this paper verifies the robustness of the improved ADRC current regulator when the inductance parameters change through simulation.

In the simulation, the demand current of q-axis is 200 A and that of d-axis is—200 A. Observe the robustness of ADRC by changing the stator inductance and compare it with PI (the parameters of PI and ADRC are well tuned, and the PI parameters are: $k_{pd} = 0.6$, $k_{id} = 40$, $k_{pq} = 0.5$, $k_{iq} = 20$; the ADRC parameters are: $\omega_d = \omega_q = 250$, $k_d = k_q = 200$, $b_d = 1618$, $b_q = 507$). The dq-axis current waveforms are shown in Figure 4. According to Figure 4, when the inductance parameters change, in terms of the peak value of dq-axis current fluctuation, the two regulators are basically equal. In terms of regulating speed, the ADRC regulator is obviously faster than the PI regulator, and there is no overshoot. The dq-axis voltage waveforms are shown in Figure 5. It can be found that when the inductance changes, the ADRC regulator has a high response speed and a small overshoot. This is because the parameters of the PI regulator are related to the inductance parameters, and the change of the inductance makes the open loop poles and zeros of the system unable

to be accurately cancelled, thus causing large current fluctuations; ADRC regards the part of parameter change as disturbance, and LESO with high gain can quickly and accurately estimate the disturbance, therefore, the adjustment speed of current and voltage is high.

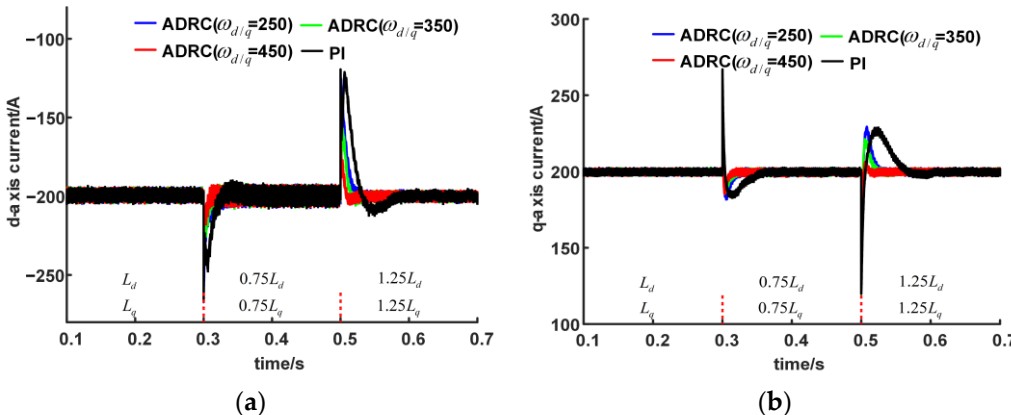

**Figure 4.** Dq-axis current waveforms: (**a**) d-axis current; (**b**) q-axis current.

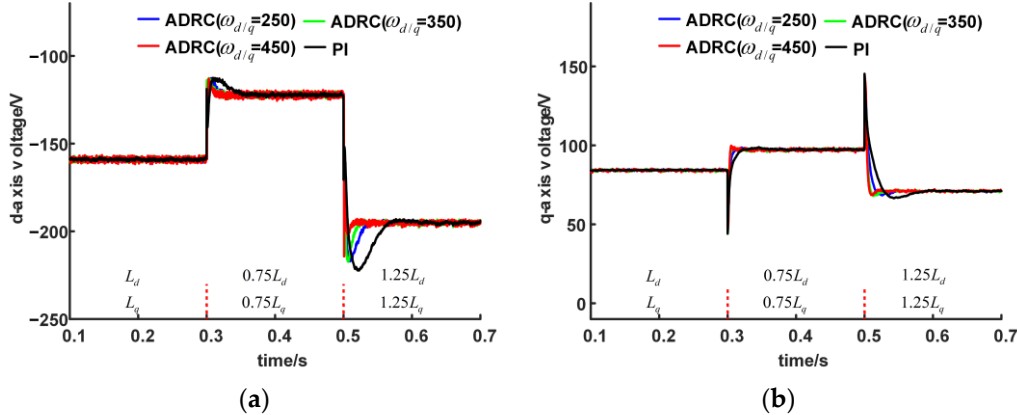

**Figure 5.** Dq-axis voltage waveforms: (**a**) d-axis voltage; (**b**) q-axis voltage.

For ADRC itself, continue to increase the value of $\omega_{d/q}$ to observe the control performance. As shown in Figures 4 and 5, the larger the $\omega_{d/q}$ is, the faster the convergence rate of LESO and the higher the observation accuracy. Therefore, with the increase of $\omega_{d/q}$, the robustness of ADRC to parameter changes is further enhanced, the current fluctuation is gradually reduced, and the adjustment speed is gradually increased, which verifies the conclusion in Section 2.2. However, by observing Figure 5, it can be seen that when $\omega_{d/q} = 450$, the output fluctuation of the regulator increases significantly, indicating that the noise suppression of LESO decreases with the increase of bandwidth.

## 4. Bench Test

### 4.1. Introduction to the Test Bench

The test bench system includes PMSM to be tested, dynamometer and DC power supply. The PMSM to be tested operates in torque (current) control mode, and the motor parameters are shown in Table 1; the dynamometer operates in the speed control mode. The physical picture of the test bench is shown in Figure 6.

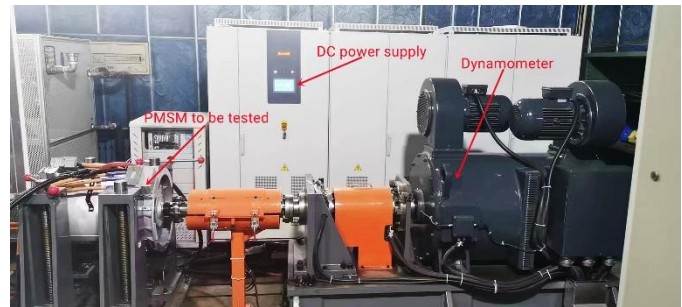

**Figure 6.** Physical picture of test bench.

*4.2. Analysis of Test Results*

4.2.1. Torque Step Test

Set the dynamometer speed to 200 r·min$^{-1}$, and test the step response of the maximum demand torque of the PMSM to be tested. Under this working condition, the demand current of q-axis is 495 A, and the demand current of d-axis is −546 A (according to the motor inductance contour diagram shown in Appendix A, the inductance parameters under this working condition are approximately $L_d = 0.522$ mH, $L_q = 1.056$ mH). Tune the parameters of PI and ADRC, respectively, until they have the same steady-state performance (the PI parameters are: $k_{pd} = 0.6$, $k_{id} = 40$, $k_{pq} = 0.5$, $k_{iq} = 20$; the ADRC parameters are: $\omega_d = \omega_q = 250$, $k_d = k_q = 200$, $b_d = 1618$, $b_q = 507$). The dq-axis current waveforms of the two current regulators are shown in Figure 7. In terms of response speed, there is little difference between the two regulators; however, in terms of overshoot, the q-axis current under the PI regulator has an approximately 7% overshoot, while the ADRC regulator can ensure no overshoot. This shows that the ADRC regulator can alleviate the contradiction between response speed and overshoot to a certain extent.

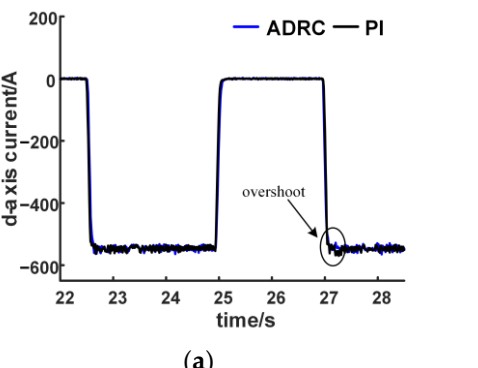
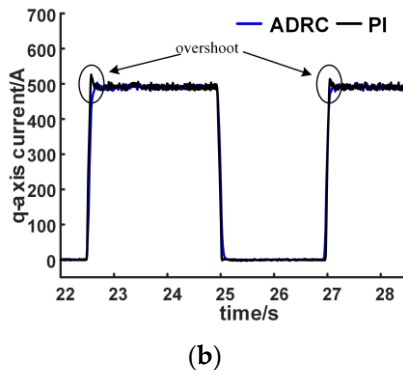

(**a**)  (**b**)

**Figure 7.** Peak torque step response(200 r·min$^{-1}$): (**a**) d-axis current; (**b**) q-axis current.

Figure 7 shows the current response when the demand current is large. The same PI and ADRC control parameters are used to analyze the working condition of the small demand current (The change rule of d-axis demand current is: 0 A, −21.8 A, −53.3 A; the change rule of q-axis demand current is: 0 A, 70 A, 119 A. According to the motor inductance contour diagram shown in Appendix A; the inductance parameters under this working condition are approximately $L_d = 0.928$ mH, $L_q = 2.522$ mH). The current response is shown in Figure 8:

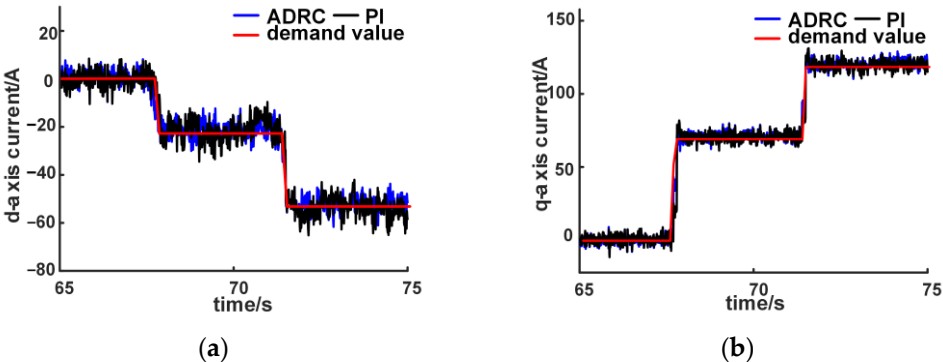

**Figure 8.** Small torque step response(200 r·min$^{-1}$): (**a**) d-axis current; (**b**) q-axis current.

According to Figure 8, it can be seen that the dynamic performance of the ADRC current regulator is good; it can maintain rapidity and has no overshoot. The q-axis current of the PI current regulator still has overshoot. However, the steady state performance of the d-axis current of the PI current regulator is poor, and there are low-frequency fluctuations.

Figures 7 and 8 can verify the robustness of ADRC when the inductance parameters change. According to the contour map of inductance, the stator inductance is smaller under high demand current conditions and larger under low demand current conditions. On the premise of not changing the control parameters, ADRC has good dynamic and steady-state responses in both conditions. However, due to parameter mismatch, the steady state performance of the PI regulator is degraded under the condition of the small demand current, which indicates that the robustness of the ADRC regulator is better than that of the PI regulator. This also indirectly proves the conclusion drawn from the simulation in Section 3.

### 4.2.2. Dynamic Test

First, the dynamometer speed is set to 200 r·min$^{-1}$, and then the speed is accelerated to 3000 r·min$^{-1}$ (PMSM peak speed) in 15 s. During this process, the motor's operating point moves along the external characteristics. As shown in Figure 9, ADRC parameters under this working condition are shown in Appendix B.

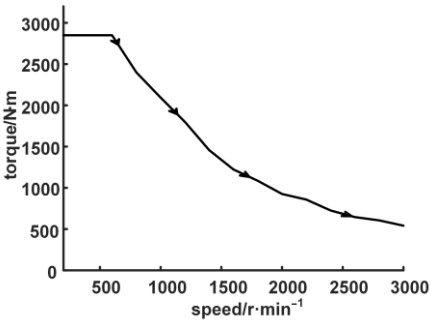

**Figure 9.** PMSM external characteristic curve.

The voltage vector mode length and dq-axis current waveform are shown in Figures 10 and 11. Under high torque, high speed and high dynamic conditions, the improved ADRC current regulator is accurate in the observation of dq-axis current, and can follow the demand current well. The total disturbance observation value of dq-axis is shown in Figure 12. It can be seen that the total disturbance of dq-axis current loop changes dramatically due to the changes of current, speed and the internal parameters of the motor, which indicates that the improved ADRC has good robustness when the load and parameters change.

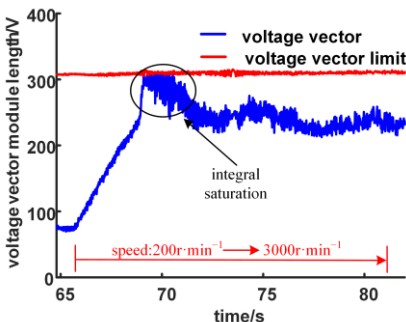

**Figure 10.** Voltage vector amplitude.

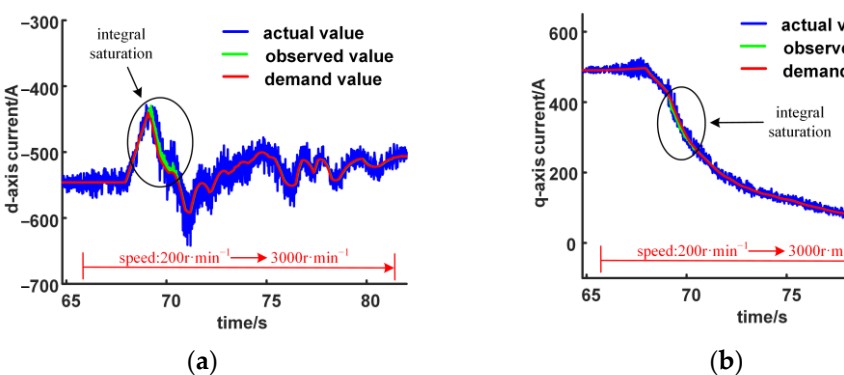

**Figure 11.** Dq-axis current waveforms: (**a**) d-axis current; (**b**) q-axis current.

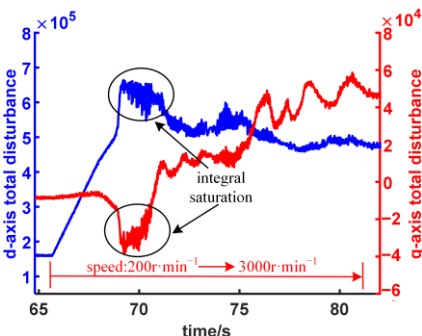

**Figure 12.** Total disturbance observed value.

In addition, the performance of anti-windup measure designed in this paper is also verified under this working condition. As shown in Figure 10, when the PMSM speed rises to about 800 r·min$^{-1}$, the amplitude of the demand voltage vector reaches the limit value of the voltage vector modulus length, which indicates that the regulator will have integral saturation. At this time, anti-windup measures play a role, and the error before and after amplitude limiting is eliminated through negative feedback regulation. The module length of the voltage vector can be limited below the limit value, which ensures that the dq-axis current follows the demand value to the maximum extent, without generating excessive current error or causing the system to be out of control. Since the anti-windup measure only works when integral saturation occurs, it is equivalent to that when integral saturation occurs; the control structure of ADRC changes, which causes fluctuations in the observed values of dq-axis current and total disturbance, thus causing some fluctuations in the demand voltage vector.

## 5. Discussion

In this paper, ADRC technology is applied to the current closed-loop control of electric vehicle PMSM, and the traditional ADRC regulator is improved from three aspects: observation error compensation, model information utilization and anti integral saturation function; the simulation results show that the improved ADRC current regulator has good robustness when the motor parameters change; after that, the bench tests are carried out. The torque step test shows that the improved ADRC current regulator has a fast step response performance without overshoot. Finally, the robustness of the improved ADRC when the load changes and the motor parameters change is tested under high dynamic conditions, as well as the effectiveness of the anti-windup measures. These studies can provide reference for other ADRC technology users.

Compared with the existing research, the highlights of this paper are:

(1) Compared with reference [11], the proposed algorithm in this paper has fewer parameters to be tuned, less computational effort, and is more suitable for engineering application;

(2) Compared with [12,14], this paper pays more attention to the robustness of the current regulator when the motor parameters change. First, the robustness of the ADRC current regulator when the motor parameters change during operation is verified through simulation, and through bench experiments, a variety of working conditions are designed for this verification;

(3) Compared with the literature [11,13], this paper has different verification methods for robustness when motor parameters change. In this paper, the compensation factor parameters are fixed, and the robustness in case of parameter mismatch is verified by changing working conditions;

(4) In order to improve the safety of the algorithm under extreme operating conditions, the anti-windup measures of ADRC are also designed and bench tested under high dynamic conditions, which has not been found in the existing research on ADRC as PMSM current regulator.

As a PMSM current regulator, ADRC also has the following limitations:

(1) Although according to the literature [16], the two gains of LESO can be expressed in the form of observer bandwidth, which reduces the number of parameters to a certain extent. However, compared with PI current regulator, ADRC still has more parameters to be tuned, which limits the application of ADRC to a certain extent. Finding a simpler parameter tuning method is the focus of the next research;

(2) When ADRC is applied to some occasions with high system bandwidth (such as high-speed motors), in order to obtain faster convergence speed and higher observation accuracy, the bandwidth of LESO will inevitably be tuned to a larger value, which will lead to the reduction of noise suppression ability. Therefore, how to reduce the noise impact when the LESO gain is large will be the focus of future research.

## 6. Conclusions

In this paper, ADRC current regulator of vehicle PMSM is designed, analyzed and improved. The following conclusions are drawn:

(1) The performance of LESO is analyzed using the frequency domain method, and the traditional ADRC algorithm is improved in three aspects: observation error compensation, model information utilization and anti windup;

(2) The simulation results show that the improved ADRC current regulator is more robust than the PI current regulator when the parameters change;

(3) The bench test results show that the improved ADRC current regulator has a fast step response without overshoot, good tracking performance and robustness when the load changes and the parameters change. In addition, the anti-windup performance is also verified.

**Author Contributions:** Conceptualization, J.W. and H.L.; methodology, J.W. and H.L.; software, J.W. and Q.M.; validation, J.W. and L.S.; formal analysis, X.Z.; investigation, D.G.; resources, Q.M. and D.G.; data curation, H.L.; writing—original draft preparation, J.W.; writing—review and editing, X.Z.; visualization, X.Z.; supervision, X.Z.; project administration, H.L.; funding acquisition, H.L. All authors have read and agreed to the published version of the manuscript.

**Funding:** This research was funded by Tsinghua University-Weichai Power Intelligent Manufacturing Joint Research Institute (JIIM04).

**Data Availability Statement:** Not applicable.

**Conflicts of Interest:** Qiang Miao and Lipeng Sun are employees of Electric Control Research Institute, Weichai Power Co., Ltd. The paper reflects the views of the scientists and not the company.

## Appendix A

Figure A1 shows the change of stator inductance with dq axis current as a reference for the change of motor stator inductance parameters with operating conditions. The contour map is given by PMSM manufacturer through finite element calculation.

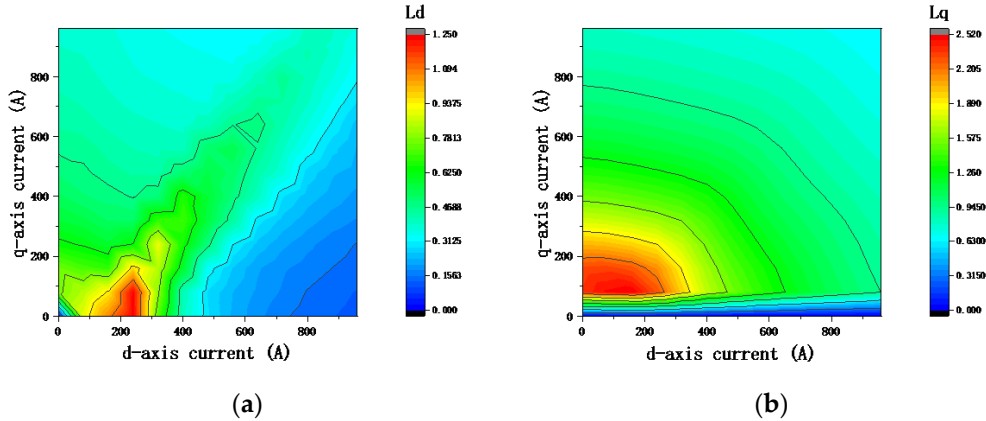

(**a**)                                                                (**b**)

**Figure A1.** Dq-axis inductance contour map: (**a**) d-axis inductance; (**b**) q-axis inductance.

## Appendix B

**Table A1.** ADRC parameters.

| Speed/r·min$^{-1}$ | $\omega_d$ | $\omega_q$ | $k_d$ | $k_q$ | $b_d$ | $b_q$ |
|---|---|---|---|---|---|---|
| 0 | 250 | 250 | 200 | 200 | | |
| 200 | 250 | 250 | 200 | 200 | | |
| 400 | 250 | 250 | 200 | 200 | | |
| 600 | 250 | 250 | 200 | 200 | | |
| 800 | 400 | 400 | 200 | 200 | | |
| 1000 | 450 | 450 | 200 | 200 | | |
| 1200 | 500 | 500 | 200 | 200 | | |
| 1400 | 500 | 500 | 200 | 200 | 1618 | 507 |
| 1600 | 500 | 500 | 100 | 100 | | |
| 1800 | 500 | 500 | 100 | 100 | | |
| 2000 | 500 | 500 | 50 | 50 | | |
| 2200 | 500 | 500 | 50 | 50 | | |
| 2400 | 500 | 500 | 50 | 50 | | |
| 2600 | 500 | 500 | 50 | 50 | | |
| 2800 | 650 | 650 | 25 | 25 | | |
| 3000 | 650 | 650 | 25 | 25 | | |

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
