# Peer review of "Current Control Method of Vehicle Permanent Magnet Synchronous Motor Based on Active Disturbance Rejection Control"

_wevj, doi:10.3390/wevj14010002_

Round 1
Reviewer 1 Report
The Authors described studies concerning on modeling of Control Method of Vehicle Permanent Magnet Syn- 2 chronous Motor Based on Active Disturbance Rejection Control. The topic is extremely important nowadays. The manuscript could be published in WEJV after revision. Below, several aspects have mentioned, which should be corrected and some doubts should be explained.
- Please improve the discussion. I suggest to compare results delivered by studies to those from literature.
Author Response
Response to Reviewer 1 Comments
Dear reviewer
Thank the reviewer for your affirmation of the topic and research direction of this paper. As the reviewer said, it is of great significance to find a control algorithm similar to ADRC to further improve the control robustness of vehicle motor system for vehicle performance. The following is the reply to the suggestions given by the reviewer:
Point 1: Please improve the discussion. I suggest to compare results delivered by studies to those from literature.
Response 1: We would like to express my heartfelt thanks again for the reviewer’s constructive comments on this article. We agree and have updated the introduction and discussion of the manuscript. We think that the difference between this paper and the literature is that ADRC current regulator is designed to be more convenient for engineering application, and the simulation and experimental verification are carried out under different working conditions for the stator inductance parameter change during motor operation. The corresponding updates can be found in the second paragraph of the introduction section and the discussion section.

Reviewer 2 Report
In this study, an improved active disturbance rejection control was proposed to replace PI current regulators in PMSM control. Some suggestions can be considered.
1. The derivation process of the proposed ADRC can be more elaborated in Section 2.1 and 2.2.
2. In section 3 and 4, please provide the results of different wdq values' influence on the output performance.
3. What are the parameters for the tuned PI controller in Section 3&4?
4. In Section 4, please also provide the experimental results as the process shown in Fig4&5.
Author Response
Response to Reviewer 2 Comments
Dear reviewer
Thank the reviewer for reviewing the manuscript and putting forward constructive suggestions. The comments of reviewer are of great significance for us to improve the quality of manuscripts. Next, we will reply to the reviewer's comments one by one and update our manuscripts.
Point 1: The derivation process of the proposed ADRC can be more elaborated in Section 2.1 and 2.2.
Response 1: The reviewer opinion is very important, and the detailed theoretical derivation process will help readers understand and reproduce the proposed content of the manuscript. We have described the derivation process of the proposed ADRC in further detail and explained the variables in the formula. The reviewer can find the corresponding content in sections 2.1 and 2.2 of the manuscript.
Point 2: In section 3 and 4, please provide the results of different wdq values' influence on the output performance.
Response 2: Thank reviewer for this constructive comment. The influence of ADRC parameters on control performance is very important to the parameter tuning of the algorithm. We agree and have updated in the manuscript. In the section 3, we add several groups of simulation data to verify the impact of different observer bandwidth on control performance. The basic rule is that the larger the observer bandwidth, the better the robustness of ADRC. However, according to the fluctuation of the current regulator output (that is, the dq-axis demand voltage) in Figure 5, the increase of the observer bandwidth will increase the ripple of the regulator output. The above modifications can be found in Section 3.2. In the experiment, the measurement noise is inevitable. In order to achieve better dynamic performance and robustness, and minimize the harmonic content of the current, we selected a better ADRC observer bandwidth for bench testing based on simulation, and calibrated the ADRC observer bandwidth parameters in the full speed range. The ADRC parameters table can be found in Appendix 2.
Point 3: What are the parameters for the tuned PI controller in Section 3&4?
Response 3: We thank the reviewer for pointing this out. We updated the manuscript and added PI parameters under each working condition in Section 3 and Section 4. It is worth mentioning that the PI parameters used in simulation and bench test are well tuned during bench test.
Point 4: In Section 4, please also provide the experimental results as the process shown in Fig4&5.
Response 4: We agreed with the reviewer's comments and updated the manuscript. We will explain our modification idea in detail below: in the simulation of section 3, we change the parameters of the motor stator inductance under the same working conditions, so that we can compare the robustness of ADRC and PI without interference from other factors. However, in the experiment, we can not achieve accurate and quantitative changes in motor parameters, so we adopted another experimental method to verify the robustness of ADRC and PI when motor parameters change. Figure 7 shows the current step response when the demand current value is the peak value (according to the inductance saturation effect, the stator inductance at this time is greater than the rated value). Under this working condition, we tune the parameters of PI and ADRC, It can be found that the steady-state performance of ADRC and PI has little difference, and the overshoot of ADRC is less than PI. Keep the control parameters unchanged, and then conduct the test under the small current demand condition (according to the inductance saturation effect, the stator inductance at this time is greater than the rated value), it can be clearly found that the steady state performance of the d-axis current of the PI current regulator has deteriorated, with low-frequency fluctuations. We believe that Figures 7 and 8 can indirectly prove that ADRC is more robust than PI. In order to prove our point of view, we give the inductance contour map given by the motor manufacturer in the form of appendix.
In addition, in the bench test under high dynamic conditions, we always keep the compensation factor parameter( bd and bq)related to the controlled object model as the reciprocal of the rated inductance value, which can also prove the robustness of ADRC when the motor parameters and load change.

Reviewer 3 Report
In this paper, ADRC technology is applied to the current closed-loop control of EV PMSM. The simulation results show that the improved ADRC current regulator has good robustness when the motor parameters change.
My comments are as follows:
1- The state-of-the-art of Control Methods of the Vehicle's Permanent Magnet Synchronous Motor are not addressed well in the introduction section, the introduction could be still improved.
2- Please pay attention when you are using the subscript in the whole presented mathematical equations
3- More experimental results are suggested to be provided and compared with the simulation results.
4- A detailed comparison with previous research work should be presented.
5- Discussion on the limitation of the proposed technique should be added.
Author Response
Response to Reviewer 3 Comments
Dear reviewer,
Thank the reviewer for reviewing this manuscript and these constructive comments, which are very important for us to improve the quality of the manuscript. Next, we will answer the comments and questions raised by the reviewers one by one:
Point 1: The state-of-the-art of Control Methods of the Vehicle's Permanent Magnet Synchronous Motor are not addressed well in the introduction section, the introduction could be still improved.
Response 1: We thank the reviewer for pointing this out. As mentioned by the reviewer, with the development of control engineering, many new technologies have been applied to PMSM control, such as model predictive control (MPC), internal model control (IMC) and model reference adaptive system (MRAS). Many scholars have also proved that these new technologies can improve the performance of motor control systems. We updated the introduction part of the manuscript to briefly introduce these new technologies.
Point 2: Please pay attention when you are using the subscript in the whole presented mathematical equations
Response 2: Thank the reviewer for pointing out the errors in the formula in the manuscript. We have carefully checked all formulas in the manuscript to ensure that they are correct, and explained the variables in the formula in the text.
Point 3: More experimental results are suggested to be provided and compared with the simulation results.
Response 3: Thank the reviewer for their constructive comments. We agree and have updated the manuscript. In Section 4.2.1, we added some experimental results to verify the robustness of ADRC regulator. In the simulation in section 3, in order to avoid the influence of other factors, we used the method of changing only the inductance parameters of the motor under fixed working conditions to verify the robustness of the proposed ADRC current regulator and the traditional PI current regulator. However, in the bench test, we were unable to accurately and quantitatively change the inductance parameters of the motor. According to the saturation effect of inductance, the inductance parameters of the motor change with the change of dq-axis current. Therefore, in Section 4.2.1, we change the inductance parameters of the motor by changing the motor demand current. First, tune the parameters of ADRC and PI under the peak current condition (at this time, the inductance parameter should be less than the rated value), and get the current waveform as shown in Figure 7. At this time, the steady-state performance of ADRC and PI has little difference, and ADRC has less overshoot; Then, keep the control parameters unchanged, carry out the small demand current working condition experiment, and get the current waveform as shown in Figure 8. At this time, it can be found that ADRC can still maintain good steady-state performance and fast dynamic performance without overshoot, while the d-axis current waveform of PI has some low-frequency fluctuations, and the q-axis current still has a certain overshoot, which shows that ADRC regulators are more robust than PI regulators when the inductance parameters change. We believe that the experimental results shown in Section 4.2.1 can also indirectly prove the correctness of the simulation results in Section 3. In addition, in order to facilitate the analysis, we also provided the contour map of the motor stator inductance changing with dq-axis current given by the motor manufacturer in Appendix 1, which is calculated based on the finite element method and can be used as a reference to judge the inductance parameters changing with the working conditions.
Point 4: A detailed comparison with previous research work should be presented.
Response 4: We agreed with the reviewer's comments and updated the manuscript. We updated the introduction part, added the introduction of the cited literature, and added the comparison with the cited literature in the discussion part. We believe that the proposed ADRC current regulator has the following outstanding points compared with the existing research:
(1) It is convenient for engineering application: compared with the literature [11], we have improved on the basis of traditional linear ADRC, without adding too many parameters to be tuned, and without involving nonlinear functions, so the calculation amount of the algorithm is small.
(2) Emphasis is placed on the verification of robustness when parameters change: many studies only verify the ADRC current regulator with respect to load disturbance (such as [12] and [14]), while we pay more attention to the impact of parameter changes (mainly stator inductance parameters) on the current regulator when the motor is running, and have designed simulations and experiments under different working conditions for verification.
(3) Design and verification of anti-windup measures. Under high speed, high torque conditions with high back EMF, the output of the current regulator may exceed the software limit amplitude. At this time, the system is in the integral saturation state, and the control performance will deteriorate. The design of appropriate anti-windup measures can ensure the robustness of the motor when operating under extreme operating conditions. At present, in the existing research on ADRC as a PMSM current regulator, there is almost no analysis of anti-windup measures.
Point 5: Discussion on the limitation of the proposed technique should be added.
Response 5: We agreed with the reviewer's comments. As the reviewer mentioned, the ADRC current regulator proposed in this paper still has some shortcomings, which are also the direction of our follow-up research. We have added a discussion on the proposed technical limitations in the discussion section.There are two main points:
(1) Although we have related the two parameters of LESO through the method proposed in literature [16], the parameter to be tuned of ADRC current regulator is still one more than that of PI regulator, which may increase the difficulty of parameter tuning. Therefore, further finding a more simple parameter tuning method is the focus of the next research.
(2) It is mentioned that the bandwidth parameter of LESO should take into account both control performance and high-frequency noise suppression. However, when ADRC is applied to some high bandwidth fields (such as high-speed motors), the bandwidth of LESO will inevitably be tuned to a large value to obtain higher convergence speed and higher observation accuracy. At this time, it may be difficult to give consideration to noise suppression. Therefore, how to optimize the structure and parameters of LESO to mitigate the contradiction between control performance and noise is also one of the future research directions.

Round 2
Reviewer 2 Report
Thank you for tackling all my questions. There is no further question.
Reviewer 3 Report
After revision, the author answered all questions and the paper can be published.